# Maternal lipid mobilization is essential for embryonic development in the malaria vector *Anopheles gambiae*

**Maurice A. Itoe** [1], **W. Robert Shaw**[1,2], **Iryna Stryapunina**[1], **Charles Vidoudez**[3], **Duo Peng**[1¤], **Esrah W. Du**[1], **Tasneem A. Rinvee**[1], **Naresh Singh**[1], **Yan Yan**[1], **Oleksandr Hulai**[1], **Kate E. Thornburg**[1,2], **Flaminia Catteruccia** [1,2]*

1 Department of Immunology and Infectious Diseases, Harvard T.H. Chan School of Public Health, Boston, Massachusetts, United States of America, 2 Howard Hughes Medical Institute, Boston, Massachusetts, United States of America, 3 Harvard Center for Mass Spectrometry, Cambridge, Massachusetts, United States of America

¤ Current address: Chan Zuckerberg Biohub Network, San Francisco, California, United States of America
* fcatter@hsph.harvard.edu

**Data Availability Statement:** The authors confirm that all data underlying the findings are fully available without restriction. The numerical data are accessible from the Harvard Dataverse online

## Abstract

Lipid metabolism is an essential component in reproductive physiology. While lipid mobilization has been implicated in the growth of *Plasmodium falciparum* malaria parasites in their *Anopheles* vectors, the role of this process in the reproductive biology of these mosquitoes remains elusive. Here, we show that impairing lipolysis in *Anopheles gambiae*, the major malaria vector, leads to embryonic lethality. Embryos derived from females in which we silenced the triglyceride lipase Ag*TL2* or the lipid storage droplet Ag*LSD1* develop normally during early embryogenesis but fail to hatch due to severely impaired metabolism. Embryonic lethality is efficiently recapitulated by exposing adult females to broad-spectrum lipase inhibitors prior to blood feeding, unveiling lipolysis as a potential target for inducing mosquito sterility. Our findings provide mechanistic insights into the importance of maternal lipid mobilization in embryonic health that may inform studies on human reproduction.

## Introduction

Reproduction is a highly energy-demanding process that requires tight metabolic control coordinated by hormones, metabolites, and immune factors [1,2]. In mammals, lipid stores in the form of triglycerides (TAGs) are deposited in the cells of adipose tissue, where they serve as a dynamic energy reservoir. As TAGs are unable to cross cell membranes, for transport to other tissues, they need to be either incorporated into vesicles or remobilized. Regulation of these processes is critically important during gestation as they coordinate the necessary energy supply to the growing fetus, and their perturbations can have negative effects on pregnancy outcomes [3,4]. In reproductive organs, lipid mobilization allows essential reproductive functions such as the development of ovarian follicles and the synthesis of steroid hormones, which prepare the endometrium for implantation [3].

repository using the link https://doi.org/10.7910/DVN/ULTW1K. Metabolomics and Lipidomics data are accessible from the EMBL-EBI MetaboLights online database using the link https://www.ebi.ac.uk/metabolights/MTBLS9881. The RNAseq data are accessible from the Gene Expression Omnibus (GEO) online database using the link https://www.ncbi.nlm.nih.gov/geo/query/acc.cgi?acc=GSE263712.

**Funding:** F.C. is funded by Howard Hughes Medical Institute (HHMI, www.hhmi.org) as an HHMI Investigator and by the National Institute of Health (NIH) grants R01AI148646, R01AI153404. M.A.I. was funded by the Charles A. King Trust postdoctoral research fellowship in Basic Sciences from Health Resources in Action (HRiA, www.hria.org). I.S. was funded by Natural Sciences and Engineering Research Council of Canada (NSERC, www.nserc-crsng.gc.ca) as a scholarship recipient (PGSD3 - 545866 – 2020). The funders were not involved in study design, data collection, analysis, or interpretation.

**Competing interests:** A patent application covering the concept of the application of lipase inhibitors to induce sterility in mosquitoes is being processed on behalf of F.C. and M.A.I. by the President and Fellows of Harvard University. The authors state that they have no other competing interests.

**Abbreviations:** ACTH, adrenocorticotropic hormone; AKH, adipokinetic hormone; ATGL, adipose triglyceride lipase; BSA, bovine serum albumin; cAMP, cyclic adenosine monophosphate; CE, cholesterol ester; DAG, diglyceride; FDR, false discovery rate; HSL, hormone-sensitive lipase; LD, lipid droplet; LPE, lysophosphatidylethanolamine; MB, maleate buffer; MPA, mobile phase A; MPB, mobile phase B; PBM, post blood meal; PBS, phosphate-buffered saline; PFA, paraformaldehyde; PKA, protein kinase A; TAG, triglyceride; TCA, tricarboxylic acid; TEM, transmission electron microscopy.

In simplified terms (reviewed in [5]), TAG remobilization follows a hydrolytic process requiring their breakdown into free fatty acids (FFAs) and glycerol, regulated by hormonal stimuli that activate lipolytic enzymes like hormone-sensitive lipase (HSL) and adipose triglyceride lipase (ATGL) [6–8]. The canonical pathway for lipolysis activation in adipocytes is driven by catecholamines, adrenocorticotropic hormone (ACTH), and secretin, which upon binding to β-adrenergic G-protein-coupled receptors stimulate cyclic adenosine monophosphate (cAMP) synthesis, activating protein kinase A (PKA). PKA activation causes the phosphorylation of HSL, which induces its translocation onto the lipid droplet (LD) surface aided by PKA-mediated phosphorylation of perilipin-1 [9,10]. Perilipin-1 phosphorylation also results in the dissociation of the co-activator comparative gene identification-58 (CGI-58) [11,12], which can then bind to ATGL and activate lipolysis [13]. Upon completion of hydrolysis, FFAs will bind to carrier proteins that transport them to target tissues in the body for their use in energy production through ß-oxidation, hormone synthesis, or other cellular metabolic needs [14].

Although the last common ancestor of insects and mammals is estimated to have lived over 500 million years ago, insects possess a similar metabolic cascade to mobilize lipids. Lipolysis of TAGs is hormonally stimulated by a neuropeptide hormone, adipokinetic hormone (AKH), that has an analogous role to glucagon in mammals and is released from the corpora cardiaca [15,16]. Once released, AKH binds to its G protein-coupled receptor (AKHR) triggering an increase in intracellular calcium levels and cAMP synthesis, followed by PKA activation [17]. Activated PKA in turn phosphorylates lipid storage droplet protein 1 (LSD1), a homologue of perilipin 1-like protein, on the surface of LDs, ultimately inducing the activation of TAG lipases to catalyze the rate-limiting step in the hydrolysis of fatty acids from TAGs to generate FFA and diglycerides (DAGs) [18–22]. In hematophagous insects, after blood feeding, complex lipids such as TAGs and phospholipids are broken down into FFAs in the midgut lumen for transport across the luminal membranes [23], where they are reutilized to synthesize DAGs, TAGs, esters, and phospholipids [24]. These lipids are then loaded—in concert with lipid transfer proteins—onto circulating lipoproteins, principally lipophorin (Lp), on the basal side of the midgut epithelium for delivery to key tissues such as flight muscles and ovaries, or directed to the fat body for storage or further mobilization [19,20,25,26]. Dynamic changes occur in the lipid profile of the fat body during a reproductive cycle [27], with substantial proportions of stored lipids being mobilized for egg production [28,29]. Lp-transported lipids, as well as the yolk protein vitellogenin (Vg), are incorporated into developing oocytes [30,31] whereupon lipid is organized into LDs and Vg crystallizes into vitellin yolk granules [32] to be degraded after oviposition to nourish the developing embryo [33]. Coordinated lipolysis regulation is mediated by multiple conserved signaling pathways, including 20-hydroxyecdysone [34,35], Hedgehog [36], Decapentaplegic [37], NF-κB [38], as well as the neuropeptides FMRFamide [39] and AKH [40]. The components of the lipolytic machinery are shared by important insect vectors of human diseases including malaria, Chagas disease, and African Trypanosomiasis [23,41–44], and could potentially be targeted to reduce the size of vector populations and aid in disease control efforts. Orthologues of TAG lipase and AKHR are significantly increased after blood feeding in the fat body of *Aedes aegypti* mosquitoes [34,40], and TAG stores in the fat body, which are sharply reduced after blood feeding due to their mobilization and incorporation into developing eggs, are not mobilized when the AKH receptor AKHR is silenced by RNAi [40]. Similarly, disruption of lipid mobilization through targeting AKH signaling, ATGL or Perilipin/LSD1 in several insects decreases fecundity [43,45] and causes embryonic lethality [21,46]. However, information regarding lipid mobilization in *Anopheles* is relatively scarce.

In one of the most important vectors of human malaria, the *Anopheles gambiae* mosquito, lipid transport by Lp has been shown to impact the development of *Plasmodium falciparum*

and *Plasmodium berghei*, respectively, a human and rodent malaria parasite. Indeed, silencing *Lp* before an infectious blood meal reduces oocyst numbers in both parasite species [47–49], while it impairs oocyst growth and sporozoite infectivity in *P. berghei* but not *P. falciparum* [48–50]. These studies also revealed that Lp, and to a lesser extent Vg, which transports approximately 5% of lipids within oocytes [51], are essential for egg development as their silencing drastically reduces the number of developing ovarian oocytes [31,47,48,50]. A role of upstream lipolytic pathways in the reproduction of this or other human malaria vectors has yet to be determined, and their function has only previously been studied in the context of pathogen infections. Silencing of the TAG lipase Ag*TL2* (a pancreatic lipase that is the functional homologue of ATGL and predominantly induced in the fat body after blood feeding [52]) by RNAi leads to accelerated *P. falciparum* oocyst growth, possibly due to the accumulation of TAGs, DAGs, and phosphatidylcholine observed in midguts and fat bodies [48], while depleting AKH1 and AKHR leads to reduced number of sporozoites in the salivary glands [53]. Whether impairing lipolysis impacts reproduction in this anopheline is unknown, and addressing this question could uncover key reproductive targets to reduce mosquito populations and curb malaria incidence in disease endemic countries.

Here, we demonstrate a key role of the *An. gambiae* maternal lipolytic machinery in shaping the development and survival of progeny. Targeted depletion of Ag*TL2* in mothers induces considerable perturbations to energy metabolism in the progeny, preventing embryos from hatching and causing embryonic lethality. Treating adult females with orlistat, a broad-spectrum hydrolase inhibitor, mimics the effects of Ag*TL2* silencing, causing dose-dependent embryonic mortality without, however, accelerating *P. falciparum* development. These data provide a proof of principle that targeting lipolysis can aid mosquito control and increase our understanding of the importance of lipid metabolism during pregnancy and embryonic development.

## Results

### Maternal triglyceride (TAG) lipolysis is critical for the survival of *An. gambiae* embryos

To determine the role of TAG mobilization in *An. gambiae* reproduction, we silenced the expression of the TAG lipase Ag*TL2* and found that its depletion reduced the number of eggs laid by females compared to controls (**Fig 1A and 1B**). Silencing of Ag*TL2*, which is induced after blood feeding predominantly in the fat body (**S1A and S1B Fig**) [52], led to an increase in glyceride levels in midguts and fat bodies after blood feeding, mostly evident at 48 hours (h) post blood meal (PBM), paralleled by a large decrease of glycerides in ovaries, where accumulation was severely impaired at both 24 h and 48 h PBM (**Fig 1C**). This effect was confirmed by reduced staining of neutral lipids in ovaries of Ag*TL2*-depleted females, while midguts and fat bodies were laden with large LDs not observed in controls (**S1D Fig**).

Embryos from Ag*TL2*-depleted mothers had a strong reduction in glyceride levels (**Fig 1D**), as expected given the low levels observed in oocytes. Strikingly, only a handful of larvae emerged from several egg batches oviposited by these females (**Fig 1E**). We confirmed that these effects were indeed due to impaired TAG lipolysis by silencing the positive regulator Ag*LSD1* (**S1C Fig**), which similarly reduced the number of eggs laid by females and induced almost complete infertility (**Fig 1F and 1G**). Embryonic development appeared comparable through gastrulation, differentiation, and into late embryogenesis in both groups (**Fig 2A**), but embryos derived from Ag*TL2*-depleted females (henceforth also referred to as dsAg*TL2* embryos) failed to hatch. Further characterization by transmission electron microscopy (TEM) confirmed normal morphology at early stages, while at later time points the gut of

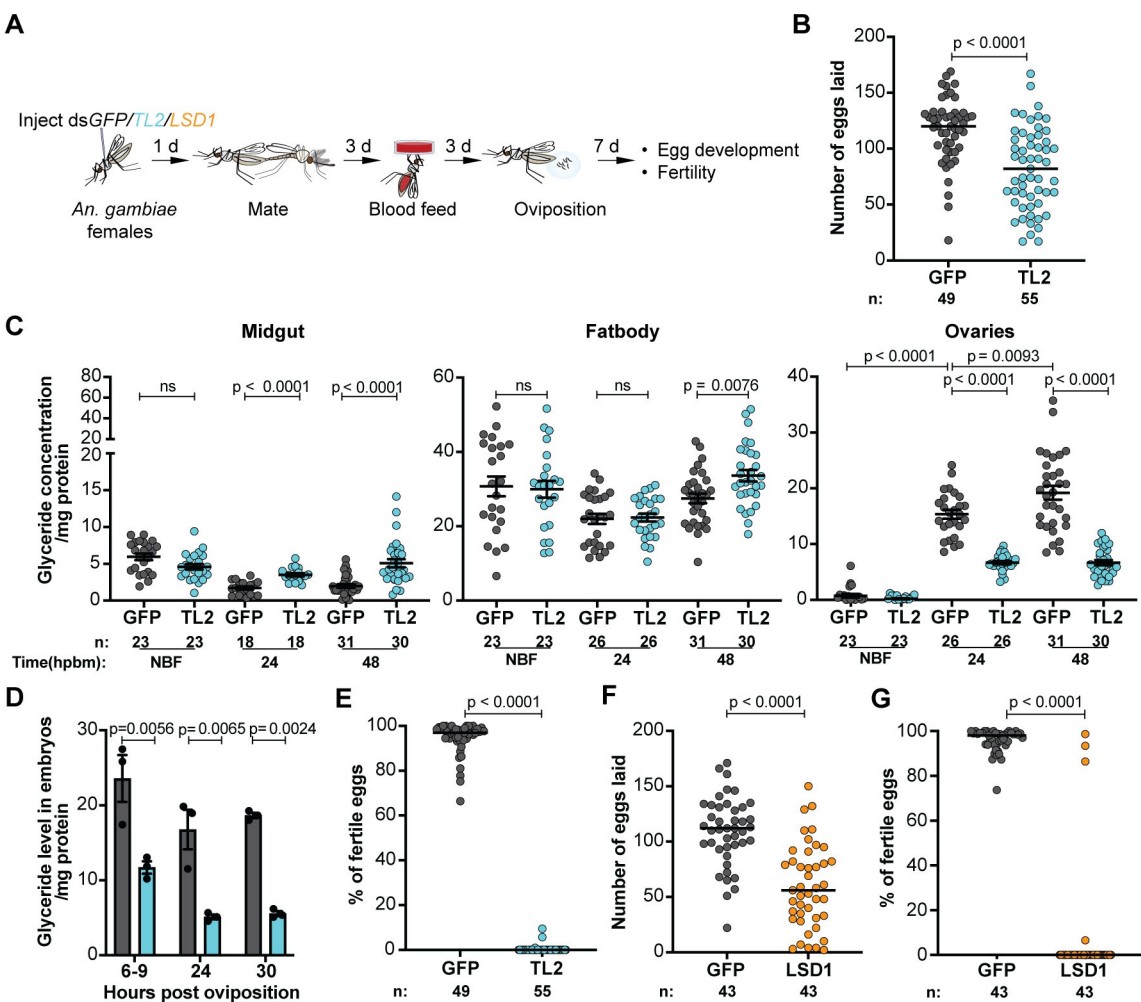

**Fig 1. Impairing lipolysis in *An. gambiae* females induces severe embryonic lethality.** (A) Schematic representation of the experimental setup, showing that females are first injected with dsAg*TL2* or control ds*GFP*, then mated and blood fed, and then analyzed for number of eggs and survival of the progeny. d = day. (B, C) Females injected with dsAg*TL2* (B) laid fewer eggs after blood feeding (Mann–Whitney test) and (C) have increased glyceride levels in midguts and fat body and decreased levels in ovaries, compared to ds*GFP*-injected controls. hPBM = hours post blood meal; NBF = non-blood fed (Least square means model, followed by FDR-corrected t test). (D) Glyceride levels are significantly reduced in embryos from AgTL2-depleted females throughout development, compared to control ds*GFP*-injected group (Ordinary 2-way ANOVA, Šídák's multiple comparisons correction). (E) Eggs oviposited by Ag*TL2*-silenced females have significantly impaired embryo survival and produce very few larvae (Mann–Whitney test). (F, G) Injection of dsAg*LSD*1 (F) significantly reduces the number of eggs laid and (G) severely decreases survival rates (Mann–Whitney test). *n* = number of mosquitoes (except for glyceride assay where n is the number of pools of tissues, including 3 insects each); 3–4 biological replicates for A–E, 2 biological replicates for F and G. Not all significant comparisons are shown, for clarity. Numerical data supporting this figure is available on the Harvard Dataverse online repository at https://doi.org/10.7910/DVN/ULTW1K.

dsAg*TL2* embryos was filled with large vesicles not observed in controls (**Fig 2B and 2C**). Altogether, these data demonstrate that the maternal lipolytic machinery is essential for embryonic development in these malaria vectors.

## Key metabolic transitions are altered in embryos from Ag*TL2*-deficient females

To reconstruct a global view of the changes induced in embryos by TAG lipase silencing in mothers, we performed multiomics analyses during embryogenesis (at 6 h, 24 h, and 38 h after

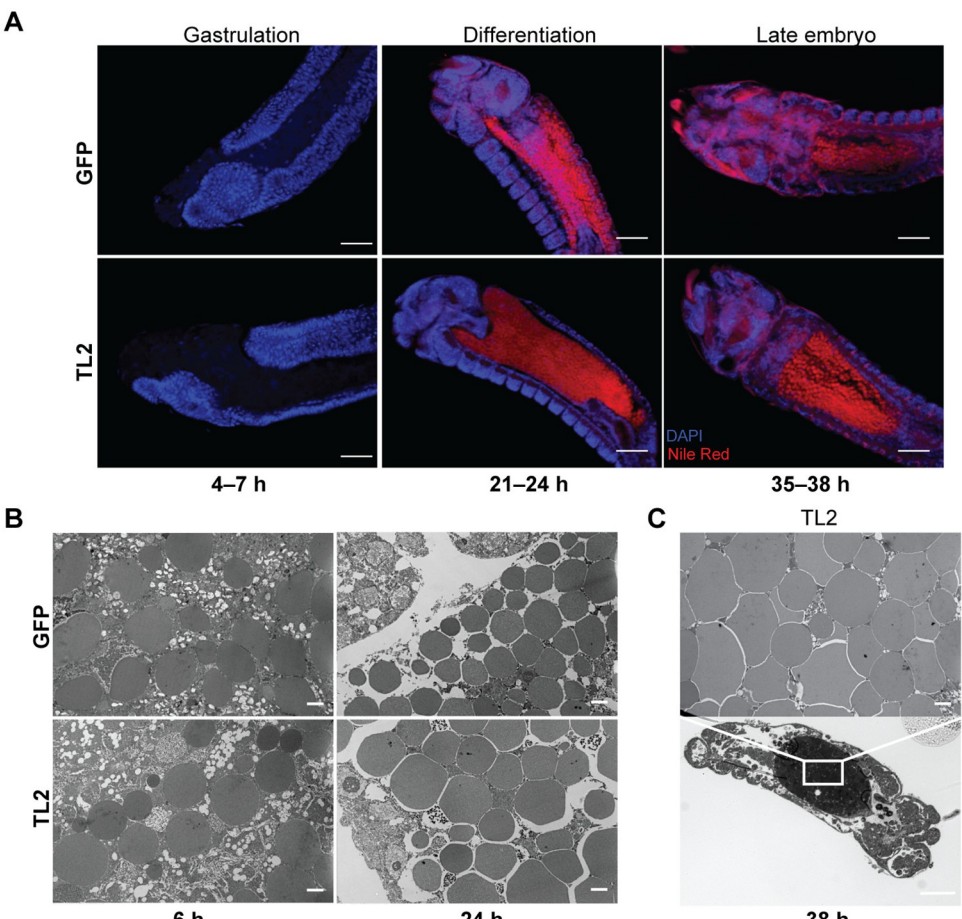

**Fig 2. Progeny of AgTL2-deficient females appear to develop normally during embryogenesis.** (A) Nile Red (neutral lipids) and DAPI (DNA) staining show apparent normal development in embryos from Ag*TL2*-knockdown and control females at 4–7, 21–24, 35–38 h after oviposition (scale bar = 50 µm). (B) TEM reveals normal granular content at early time points in guts of embryos from Ag*TL2*-knockdown females (scale bar = 2 µm). (C) Brightfield (lower panel) and TEM (upper panel) reveals large granules in the midgut of embryos from AgTL2-depleted females at 38 h after oviposition (Brightfield, scale bar = 50 µm; TEM, scale bar = 2 µm). Representative images from 3 experiments are shown. TEM, transmission electron microscopy.

oviposition), collecting lipidomics, metabolomics and RNAseq data sets. dsAg*TL2* embryos inherited significantly reduced levels of maternal lipids than controls, driven mainly by reduced TAGs (**Fig 3A and S1 Data**), consistent with our glyceride quantifications, while cholesterol ester (CE) and lysophosphatidylethanolamine (LPE) levels were increased (**Figs 3A, S2A, and S2B**). The changes in TAG and CE levels were not specific for a particular fatty acid composition (**S2C and S2D Fig**).

Additionally, embryos from AgTL2-depleted mothers showed highly different metabolic profiles (**Fig 3B**) indicative of substantial breakdown in key metabolic processes. Indeed, while in controls we observed a steady increase in major metabolites over time—including sugars (trehalose, lactose, and glucose), intermediates of the tricarboxylic acid (TCA) cycle and ß-oxidation, and nucleotides and their derivatives—dsAg*TL2* embryos had significantly reduced levels of most of these metabolites at one or more time points, while hallmarks of purine and pyrimidine catabolism (ureidopropionic acid, hypoxanthine, xanthine, and uric acid) were notably increased at 38 h post oviposition (**Figs 3C and S3 and S2 Data**).

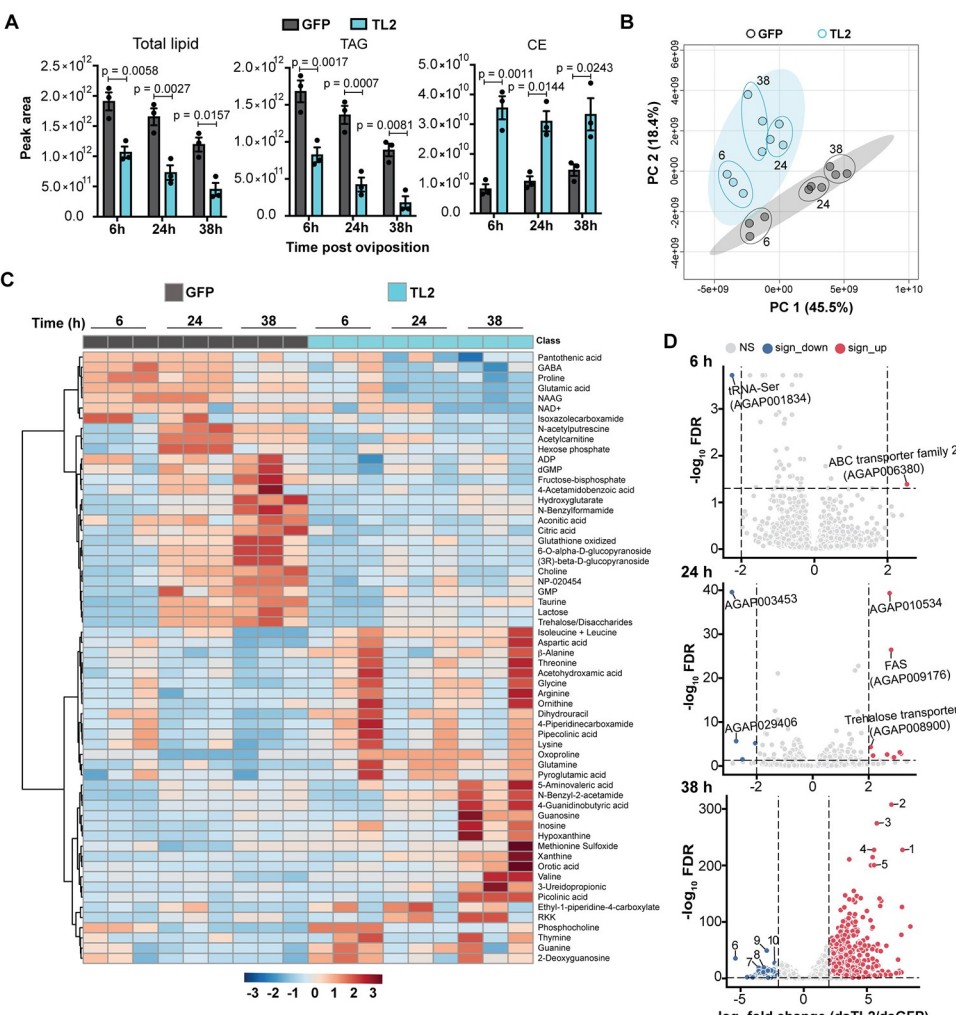

**Fig 3. AgTL2 embryos have altered lipids, metabolites, and transcriptional profiles during development.** (A) LC-MS analysis shows reduced levels of total lipids and TAGs and increased levels of CEs in embryos from AgTL2-depleted females compared to controls throughout development (Ordinary 2-way ANOVA, Šídák's multiple comparisons correction). (B) Principal component analysis shows distinct clustering of metabolites in AgTL2 embryos relative to controls. Small circles represent individual replicates. Numbers indicate time after oviposition. (C) Heatmap of metabolites analyzed by LC-MS showing levels of the 60 most highly dysregulated metabolites in AgTL2 embryos compared to controls throughout development (values of $t$ test statistic, range: blue to red = significant decrease to increase) (3 biological replicates represented by triplicate columns at each time point). (D) Volcano plots of RNAseq data of embryos from control and AgTL2-depleted females at 4–6 h (6 h), 22–24 h (24 h), and 36–38 h (38 h) after oviposition (4 biological replicates). Few genes, including FAS, ABC transporter, trehalose transporter, and tRNA-Ser, are altered at early time points, while 785 genes are up-regulated and 220 genes are down-regulated at the final time point. At 38 h, top 5 up-regulated genes (log$_2$ FC ≥ 2.0) are: 1: AGAP007650 (Growth Arrest and DNA-damage-inducible protein); 2: AGAP001610 (unknown); 3: AGAP029366 (Retrotrans-gag domain-containing protein); 4: AGAP009460 (c-Jun N-terminal Kinase JNK); 5: AGAP028615 (unknown). Top 5 down-regulated genes (log$_2$ FC ≤ −2.0) are: 6: AGAP009591 (formyltetrahydrofolate dehydrogenase); 7: AGAP003620 (unknown); 8: AGAP007903 (Excitatory amino acid transporter 2 isoform 2); 9: AGAP009896 (Proton-coupled amino acid transporter); and 10: AGAP001742 (unknown). See Tables 1 and S1 and S2 Data and S3 Data. Numerical data supporting this figure is available in the Harvard Dataverse online repository at https://doi.org/10.7910/DVN/ULTW1K. CE, cholesterol ester; FAS, fatty acid synthase; TAG, triglyceride.

While changes in lipid and metabolite composition started early during embryogenesis, we observed limited differences in the transcriptional profile of the 2 groups at the 2 earlier time points (**Fig 3D** and **S3 Data**). One exception was the up-regulation of genes involved in fatty

**Table 1. Selected representative key genes/pathways from RNAseq data that are significantly altered during embryonic development in TL2-depleted condition relative to controls.**

| Direction of change | Time point (h) | Gene ID | Biological/Molecular function | Log$_2$Fold Change | Adjusted p-values |
|---|---|---|---|---|---|
| Down-regulated | 24 | AGAP003453<br>AGAP029406<br>AGAP029964<br>AGAP012520 | Cytokinesis | -2.89<br>-2.73<br>-2.51<br>-2.05 | 2.68E-40<br>2.21E-06<br>0.03203654<br>6.56E-06 |
| Up-regulated | 24 | AGAP003582<br>AGAP007858<br>AGAP009176<br>AGAP008900 | Sugar transport and lipid synthesis | 3.15<br>3.11<br>2.80<br>2.07 | 0.00155915<br>0.00077544<br>3.69E-27<br>4.90E-05 |
| Down-regulated | 38 | AGAP009591 | Folate metabolism | -5.41 | 2.77E-36 |
| | 38 | AGAP012959<br>AGAP013072<br>AGAP013169<br>AGAP013483 | Neuropeptide signaling | -2.87<br>-2.66<br>-2.51<br>-2.41 | 7.85E-12<br>4.68E-05<br>3.31E-12<br>4.68E-11 |
| | 38 | AGAP010122<br>AGAP003383<br>AGAP010437<br>AGAP003675 | Ecdysis and cuticle formation | -3.85<br>-3.39<br>-3.35<br>-2.33 | 0.00219844<br>0.03041433<br>2.01E-14<br>4.90E-16 |
| | 38 | AGAP001123<br>AGAP012321<br>AGAP005047<br>AGAP009520<br>AGAP009857 | Sensory perception | -3.58<br>-3.57<br>-3.13<br>-2.40<br>-2.32 | 0.01244223<br>9.15E-14<br>7.69E-05<br>0.00248799<br>0.01534543 |
| Up-regulated | 38 | AGAP004583<br>AGAP007650<br>AGAP004581<br>AGAP004582<br>AGAP012891 | Protein folding<br>(Heat shock proteins) and DNA damage | 8.44<br>7.82<br>7.78<br>7.73<br>6.13 | 1.36E-92<br>3.57E-228<br>3.86E-127<br>6.74E-78<br>1.52E-06 |
| | 38 | AGAP007858<br>AGAP002969<br>AGAP009701<br>AGAP005576 | tRNA aminoacylation | 6.05<br>4.16<br>4.01<br>3.90 | 5.77E-129<br>8.84E-101<br>4.04E-111<br>9.38E-140 |
| | 38 | AGAP012154<br>AGAP012643<br>AGAP009176 | Solute transport and lipid synthesis | 4.36<br>3.95<br>3.08 | 2.19E-42<br>7.33E-77<br>9.60E-77 |

acid synthesis in dsAg*TL2* embryos at 24 h post oviposition, likely to compensate for the severe energetic deficits derived from the reduction in fatty acids and ß-oxidation (**Table 1 and S4A and S4B Fig**). However, by 38 h post oviposition, there were large differences in the dsAg*TL2* group characterized by hundreds of up- and down-regulated genes. Among the down-regulated genes, we observed a large decrease in neuropeptides and G-protein coupled receptors involved in sensory perceptions (**Table 1 and S4A, S4B, and S5 Figs**). Indeed, corazonin and eclosion hormone, 2 neuropeptides produced by neurosecretory cells in the brain that regulate ecdysis [54,55], and peptide hormones such as cchamide-1-related, secreted by endocrine cells localized in the adult gut and possibly regulating gut-brain communication [56,57], were all substantially down-regulated (**Table 1 and S4B Fig**). Several cuticular protein genes were also suppressed, suggesting inhibition of the second cuticle that is usually formed before hatching in Diptera (58). Up-regulated genes showed a strong signature of stress response and protein folding, including heat shock proteins, growth arrest and DNA-damage inducible protein, and a c-Jun N-terminal Kinase (JNK), possibly indicative of cellular death (**Table 1 and S4A and S4B Fig**).

These data reveal that maternal TAGs are essential for driving key metabolic processes during embryogenesis, such that when these lipids are limiting, embryos cannot complete

development. A summary of the major metabolic changes and how they may be connected is provided in **S5 Fig**.

## A hydrolase inhibitor induces severe embryonic lethality in *An. gambiae*

The remarkable phenotype observed when silencing Ag*TL2* and Ag*LSD1* makes TAG hydrolysis an appealing target for reducing the size of mosquito populations in field settings. Indeed, strategies that rely on chemically interfering with the reproductive biology of *Anopheles* mosquitoes are already in use in malaria endemic regions to mitigate the negative effects of widespread insecticide resistance, as in the case of insecticide-treated bed nets also containing the growth regulator pyriproxyfen [59–61], a juvenile hormone analogue that reduces the mosquito reproductive output [62]. We hypothesized that using lipase inhibitors may also induce embryonic lethality when administered to adult females before blood feeding. Consistent with our hypothesis, injections of orlistat—a broad-spectrum hydrolase inhibitor that covalently binds to Ser152 in the active site of mammalian lipases [63]—into the hemolymph of females prior to blood feeding significantly impaired egg development and induced strong embryonic lethality in a concentration-dependent manner (**Fig 4A–4C**).

In field settings, mosquitoes can absorb insecticides and sterilizing compounds via their legs when they land on bed nets incorporating these ingredients, as in the case of pyriproxyfen nets [59–62]. When we tested this method of delivery by allowing females to rest on a surface

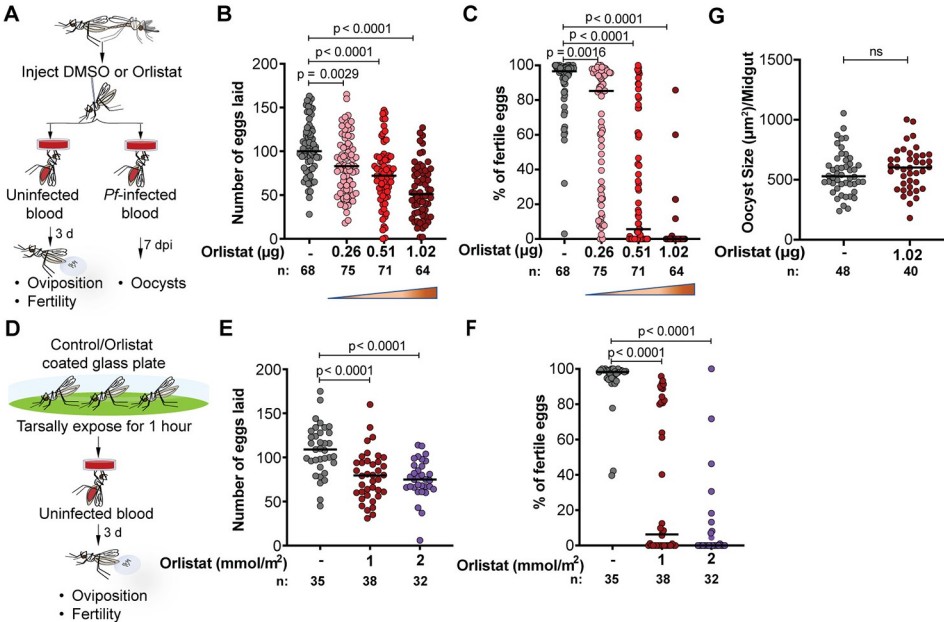

**Fig 4. Treatment with orlistat induces severe embryonic mortality but does not affect *P. falciparum* growth.** (A) Experimental scheme showing how 4-day-old adult *An. gambiae* females are injected with control (DMSO) or orlistat and then blood fed with either infected or uninfected blood. (B, C) Orlistat treatment decreases both (B) the number of eggs laid and (C) the survival of the embryos in a dose-dependent manner (4 biological replicates). (D) Experimental scheme showing how *An. gambiae* females are exposed for 1 h to orlistat-treated glass or control plates before blood feeding to measure oviposition and fertility. (E, F) Tarsal exposure to orlistat (E) decreases the number of eggs laid and (F) reduces embryo survival (2 biological replicates). (G) Orlistat treatment has no effect on *P. falciparum* oocyst size (two-tailed Unpaired *t* test) (2 biological replicates). d = day; *n* = number of individual mosquitoes; pi = post injection. (Ordinary 1-way ANOVA (B and E) and Kruskal–Wallis test (C and F), Dunn's multiple comparisons correction). Numerical data supporting this figure is available in the Harvard Dataverse online repository at https://doi.org/10.7910/DVN/ULTW1K.

coated with orlistat before blood feeding, their reproductive output was also strongly impaired in a dose-dependent manner (**Fig 4D–4F**). These data reveal that the development of *Anopheles*-specific lipase inhibitors may represent a promising tool for field interventions aimed at reducing the size of field mosquito populations.

## Orlistat does not affect *P. falciparum* development in the mosquito

Our previous work had shown that the growth of *P. falciparum* oocysts is accelerated in females depleted for AgTL2, leading to early sporozoite invasion of the salivary glands [48]. This effect would have obvious negative consequence when using lipase inhibitors to induce sterility in field populations, as it would lead to mosquitoes that can better support parasite transmission. We first confirmed that TAG lipolysis affects parasite growth in *An. gambiae* by silencing Ag*LSD1* prior to infection with *P. falciparum*. Paralleling results previously observed in Ag*TL2*-silenced females, AgLSD1 depletion accelerated oocyst growth without affecting prevalence or intensity of infection (**S6A and S6B Fig**), leading to increased sporozoite numbers at early time points (12 to 13 days) after infection relative to controls (**S6C Fig**). Orlistat treatment, however, did not induce an increase in parasite growth rates, and oocyst size was indistinguishable from controls (**Fig 4G**), providing reassurance that this strategy for inducing sterility would not have unwanted consequences on parasite transmission. Interestingly, orlistat-treated mosquitoes had elevated glyceride levels in the midgut but not in the fat body (**S6D Fig**), as instead observed in Ag*TL2*-silenced females, possibly explaining the lack of effects on oocyst growth and showing this inhibitor may preferentially affect midgut lipases rather than AgTL2. When combined, these data further support the development of *Anopheles*-specific lipase inhibitors for controlling mosquito populations that transmit malaria.

## Discussion

Our data reveal an unappreciated role for maternal TAG remobilization in shaping the development of *An. gambiae* embryos. We show that impairing lipolysis via AgTL2 depletion in the fat body, the fat storage tissue, leads to a substantial decrease in TAGs in the ovaries of adult females, the destination of lipids shuttled via the function of lipoproteins. In turn, we demonstrate that inheriting vastly reduced TAG levels triggers a cascade of nutritional dysregulation that prevents embryos from hatching and leads to their death. The almost complete penetrance of this phenotype led us to test the possibility of targeting lipolysis in mosquitoes to reduce the size of field populations. We were pleasantly surprised by the efficacy in proof of principle experiments of the lipase inhibitor orlistat, which induced significant embryonic lethality when applied to females. Orlistat is a drug used in humans to counter obesity and therefore it would not be suitable for a mosquito-targeting strategy. However, in the future, it may be feasible to generate *Anopheles*-specific inhibitors that target the function of AgTL2, AgLSD1, or other factors critical to mosquito lipolysis, for incorporation in bed nets (or other vector control strategies like indoor residual sprays [64]). Developing these inhibitors for mosquito control may prove challenging given the close homology between mosquito and human lipases [21], although it is possible that allosteric inhibitors may show higher specificity against mosquito enzymes [65]. Potential impacts on nontarget organisms would need to be evaluated through toxicological testing in mammals and insect surveys during efficacy trials of the intervention. If necessary, effects could be mitigated through usage methods, dosages, or formulations that reduce exposure of other insects to these inhibitors, such as physical incorporation into bed nets, which generally only target night-biting species contacting the net. Importantly, as mentioned above, compounds that interfere with mosquito reproduction are already incorporated into bed nets utilized in malaria control strategies. Indeed, dual-ingredient nets

including the insect growth regulator pyriproxyfen show promise in malaria endemic countries [59–61], as they act as secondary active agents in the event of mosquito resistance to the primary insecticide. This is particularly relevant at a time when insecticide-treated nets, after contributing to averting millions of malaria cases worldwide [66], are showing worrying signs of inefficacy [67], representing one of the principal causes behind the increase in malaria cases after years of decline [68,69]. Once developed, *Anopheles*-specific inhibitors of the lipolytic machinery could therefore contribute to feed the pipeline for the next generation of dual-ingredient bed net products.

Besides Lp-transported lipids, the yolk protein Vg conveys approximately 5% of total oocyte lipids [51], and so yolk deposition could potentially be affected in Ag*TL2* embryos and contribute to their abnormal development. Interestingly, embryonic lethality was reported when maternally deposited enzymes required for yolk usage in the embryo were silenced in *Rhodnius prolixus* embryos [70], suggesting that such phenotypes are due to an inability to mobilize yolk effectively. Our studies in *An. gambiae* have however shown that embryonic lethality occurs much earlier when *Vg* is silenced. Indeed, Vg depletion in mothers, which led to reduced yolk protein deposition into the oocyte, arrested embryonic development within the first few hours of oviposition [31], suggesting the late mortality observed in dsAg*TL2* embryos is not due to defects in yolk deposition. The effects of Ag*TL2* silencing on *Anopheles* reproduction are in line with phenotypes observed in insects after manipulation of other components of the lipolytic machinery. In *Drosophila*, knockout of *brummer* (*bmm*), the orthologue of the mammalian adipose TAG lipase ATGL, in both parents leads to excessive fat accumulation and impaired lipid mobilization from fat bodies, and severely affects hatching of mutant embryos by unknown mechanisms [21], and similar results were obtained in recent studies in *Bombyx mori* [46] and *R. prolixus* [45]. Although these studies across insects did not determine the mechanisms triggered by inhibition of lipolysis, they highlight a degree of conservation in the importance of TAG mobilization in insect reproduction that could well be exploited to target populations of insect vectors of human and animal pathogens, as well as agricultural pests.

Our multi-omics analyses revealed that embryos from AgTL2-depleted mothers have considerable deficits in key metabolic processes. Carbohydrate metabolism and ß-oxidation are shut down in early developmental phases, followed by disruption of the TCA cycle. While these metabolic defects are not apparent in the histological samples and are not reflected by changes in transcriptomic profiles at the early RNAseq time points, it is likely that their accumulation leads eventually to the mis-regulation of several genes observed at the last time point analyzed (38 h post oviposition), a time when control embryos are close to hatching. Up-regulated genes are enriched in stress response and protein folding genes, revealing embryos that are under severe stress. This is paralleled with a strong down-regulation in key sensory signaling receptors and neuropeptides, eclosion hormone [54,55], and cuticular proteins that form the second cuticle needed before hatching [58]. LDs are known to be multifunctional storage compartments in the developing embryo [71], providing a source of not only TAGs and phospholipids but other lipids classes as well, such as the waxy saturated hydrocarbons required for cuticle synthesis [72]. Along with a significant decrease in expression of 26 annotated cuticle-related genes (**S3 Data**), a lack of LD breakdown in dsAg*TL2* embryos may sequester these structural lipids away from cuticle synthesis enzymes, preventing the production of adequate cuticle for hatching [58]. Lipids stored within LDs may also serve as the source of several key signaling molecules, such as eicosanoids, hormones derived from polyunsaturated FFAs with roles in egg and eggshell formation [73]. Limited eicosanoid signaling may also explain embryonic lethality as eggshell defects and low hatching rates were recently reported in *R. prolixus* after *bmm* silencing [45], and eggshell defects have been observed after inhibition of eicosanoid signaling in *Aedes* mosquitoes [74].

We also observed a highly significant GO term enrichment for neural functions in down-regulated genes at 38 h post oviposition (top 4 terms), including neuropeptide and GPCR signaling pathways. CCHamide1-related proteins, induced in control but not dsAg*TL2* embryos, are arthropod neuropeptides secreted by enteroendocrine cells in the gut upon sensing of different dietary conditions in the intestinal lumen. They regulate activity of dopaminergic neurons in the brain to control gut motility and other physiological behaviors [75–77], although interestingly they have not previously been implicated in embryogenesis and hatching. Lipids and their derivatives are required for neuronal signaling through their roles in synapse function and axonal wrapping [78,79]. Indeed, we observed increased levels of LPE and decreased ceramide phosphoethanolamine (CerPE) in dsAg*TL2* embryos at 38 h post oviposition, changes that resemble *Drosophila* glial mutants unable to ensheathe axons [80], again suggestive of neuronal defects. In mammals, deficiencies in lipid metabolism and lipid signaling have been associated with developmental and cognitive problems as well as with neurodegenerative diseases. Imbalances in fatty acids during gestation causes altered brain development in the fetus, while severe fatty acid deprivation can impair cognitive functions (reviewed in [78]). Although we could not pinpoint the precise mechanisms behind embryonic lethality, our results show how reduced energy stored caused by impaired lipid remobilization and low TAG levels leads to the collapse of normal metabolic functions, eventually causing severe stress and cellular death.

Finally, our data confirm that *P. falciparum* parasites are affected by the mosquito lipolytic machinery. Depletion of AgTL2 (previous studies [48]) and AgLSD1 (this study) leads to accelerated oocyst growth rates and faster sporozoite invasion of salivary glands. Interestingly, however, such acceleration is not recapitulated by orlistat treatment, an observation that implicates fat body (rather than midgut) lipids in parasite growth. Indeed, while both AgTL2 depletion and orlistat treatment induce glyceride accumulation in the midgut (this study and [48]), differences in glyceride levels in the fat body are only detected after lipase silencing. This difference suggests that orlistat, while closely phenocopying the effects of AgTL2 depletion in terms of oogenesis and embryonic development, may inhibit different or additional aspects of the female lipolytic machinery. Regardless of the mechanism, when combined with the observed effects on fertility, these data support the idea that if *Anopheles*-specific lipase inhibitors were to be generated, they may represent promising tools for field interventions aimed at reducing the size of field populations of malaria vectors, aiding efforts to control this devastating disease.

## Materials and methods

### Rearing of *An. gambiae* mosquitoes

*Anopheles gambiae* (the standard laboratory wild-type strain G3) were reared in cages at 27°C, 80% humidity, and on a 12 h light-dark cycle. Adults were fed 10% glucose ad libitum and weekly on purchased human blood (Research Blood Components) for maintenance.

### Culturing of *P. falciparum* parasites

*Plasmodium falciparum* (NF54 strain) parasites were maintained as asexual stages in human erythrocytes and gametocytes were induced as described in [48].

### Embryo collection

Glass bowls containing circular Whatman filter papers submerged in deionized water were provided for 2 h to mated control and Ag*TL2*-knockdown females at 4 days (d) PBM. Embryos

were aged outside cages and collected for an assay of interest at 4 to 6 h, 22 to 24 h, and 36 to 38 h after oviposition.

## Measurement of triglyceride (TAG) levels

Non-blood fed and blood fed control and Ag*TL2*-silenced adult female *An*. *gambiae* mosquitoes were cooled on ice and decapitated in 1× phosphate-buffered saline (PBS). Ovaries, midgut, and fat body were dissected and transferred into separate tubes containing two 2 mm beads, diluted NP-40 substitute assay buffer (Triglyceride Assay Colorimetric kit, Cayman Chemicals), and 1× cocktail of protease inhibitor (cOmplete Protease Inhibitor Cocktail, EDTA-Free, Roche). For orlistat experiments, tissues were dissected from control or orlistat-treated mated wild-type blood fed mosquitoes at 48 h PBM. Each tube contained tissues pooled from 3 animals. For TAG determination in embryos, 200 embryos laid by control and Ag*TL2*-silenced females were collected at 6, 24, and 30 h after oviposition. TAG quantification in tissues and embryos was performed according to the manufacturer's instructions in the Triglyceride Assay Colorimetric kit (Cayman Chemicals). The assay quantifies glycerol content derived after release of fatty acids by lipoprotein lipase hydrolytic activity. Protein content in each pool was determined by Bradford assay and used for normalization of TAG levels.

## Embryo fixation, immunofluorescence assays, and transmission electron microscopy (TEM)

Embryos were fixed as described by [81]. Briefly, embryos were rinsed with distilled water onto a filter paper at the indicated time point and bleached for 75 s with 25% bleach (2% sodium hypochlorite). After rinsing 6 to 8 times with distilled water, they were transferred into scintillation vials with 1:1 solution of 9% paraformaldehyde (PFA) and heptane and rotated on a roller shaker for 25 min at 70 rpm at room temperature. The water phase was replaced once and then washed with distilled water for 30 min on a shaker. Following incubation with boiled and iced-cold distilled water for 30 s and 15 s, respectively, the water phase was removed, and heptane was replaced. To remove the endochorion, an equal volume of methanol was added, and the vials were strongly swirled once and allowed to stand for 15 min at room temperature. The heptane-methanol mixture was removed, and embryos were washed 3 to 4 times with methanol and stored in fresh methanol at −20°C until the day of removal of eggshell. To remove eggshells, embryos were transferred onto a double-sided tape and peeled according to [82]. For immunofluorescence assays, embryos were permeabilized and blocked with a 2% bovine serum albumin (BSA) in 0.1% Tween-20 in 1× PBS solution for 30 min at room temperature.

For TEM, peeled embryos were collected in 200 μl of fixative (2.5% PFA, 5% glutaraldehyde, 0.06% picric acid in 0.2 M cacodylate buffer) and submitted to the Harvard Medical School Electron Microscopy Core. There, they were washed once with 0.1 M cacodylate buffer, twice with water, and then postfixed with 1% osmium tetroxide and 1.5% potassium ferrocyanide in water for 1 h at room temperature. Samples were then washed twice in water followed by once in 50 mM maleate buffer (pH 5.15) (MB). Next, the samples were incubated for 1 h in 1% uranyl acetate in MB, followed by 1 wash in MB, and 2 washes in water. Samples were then dehydrated via an ethanol gradient: increasing concentrations of ethanol were used with samples placed successively in 50%, 70%, 90%, 100%, 100% ethanol for 10 min each. After dehydration, samples were placed in propylene oxide for 1 h and then infiltrated overnight in a 1:1 mixture of propylene oxide and TAAB 812 Resin (www.taab.co.uk, #T022). The following day, samples were embedded in TAAB 812 Resin and polymerized at 60°C for 48 h. Ultrathin sections (roughly 80 nm) were cut on a Reichert Ultracut-S microtome, sections were picked up onto

copper grids, stained with lead citrate, and imaged in a JEOL 1200EX transmission electron microscope equipped with an AMT 2K CCD camera.

## Staining of lipids with LD540 or Nile Red

Ovaries, midguts, and fat body tissues were dissected from control ds*GFP*, *TL2*-silenced, control DMSO-, or orlistat-injected females at 48 h PBM and fixed in 4% PFA for 45 min at room temperature. Tissues were washed twice with 1× PBS for 15 min and incubated in permeabilization and blocking solution (1% BSA, 0.1% Tween 20, 1× PBS) for 30 min with shaking at room temperature. Samples were stained with LD540 in 1× PBS (0.5 µg/ml) according to [83]. Nuclei were stained with 4′,6-diamidino-2-phenylindole (DAPI) in 1× PBS (1 µg/ml) for 5 min and mounted on microscope slides with Vectorshield (Vector laboratories). Peeled embryos were fixed, permeabilized, and blocked as above but stained with Nile Red (10 µg/ml) for 30 min with shaking at room temperature. Images were captured on an Inverted Zeiss microscope Observer Z1.

## Lipidomic and metabolomic analyses

All solvents were HPLC-MS grade from Sigma-Aldrich.

For lipidomic and metabolomic analyses of *An. gambiae* embryos, 200 embryos oviposited by control and AgTL2-knockdown blood fed mosquitoes in 3 biological replicates were collected at 4 to 6 h, 22 to 24 h, and 36 to 38 h after laying with 1 ml of methanol into a vial containing five 2 mm steel beads. Embryos were homogenized in a bead-beating homogenizer (Mini-Beadbeater 96, BioSpec Products) for 5 min at 2,400 rpm in a cold block and transferred to 8 ml glass vials (Duran Wheaton Kimble Life Sciences). After rinsing the bead-beater tubes with an additional 1 ml of methanol, 4 ml of cold chloroform was added to each glass vial and vortexed for 1 min, and 2 ml of ultrapure water was added to each vial and vortexed for another 1 min. Glass vials were centrifuged at 3,000 $g$ for 10 min at 4˚C, and 3 ml of the upper aqueous phase and 3 ml of the lower chloroform phase were transferred into separate glass vials for metabolomic and lipidomic analyses, respectively.

For metabolomics, samples were dried under nitrogen flow and resuspended in 50 µl of acetonitrile 25% in water. A pool was created by combining 10 µl of each sample. Samples were analyzed by LC-MS on a Vanquish LC coupled to an ID-X MS (Thermo Fisher Scientific); 5 µl of sample was injected on a ZIC-pHILIC PEEK-coated column (150 mm × 2.1 mm, 5 µm particles, maintained at 40˚C, Sigma-Aldrich). Buffer A was 20 mM ammonium carbonate, 0.1% ammonium hydroxide in water and Buffer B was acetonitrile 97% in water. The LC program was as follow: starting at 93% B, to 40% B in 19 min, then to 0% B in 9 min, maintained at 0% B for 5 min, then back to 93% B in 3 min and re-equilibrated at 93% B for 9 min. The flow rate was maintained at 0.15 ml min$^{-1}$, except for the first 30 s where the flow rate was uniformly ramped from 0.05 to 0.15 ml min$^{-1}$. Data was acquired on the ID-X in switching polarities at 120,000 resolution, with an automatic gain control target of 1e5, and an m/z range of 65 to 1,000. MS1 data was acquired in switching polarities for all samples. MS2 and MS3 data was acquired on the pool sample using the AquirX DeepScan function, with 5 reinjections, separately in positive and negative ion mode. Data was analyzed in Compound Discoverer 3.2 (Thermo Fisher Scientific). Identification was based on MS2/MS3 matching with a local mzvault library and corresponding retention time built with pure standards (level 1) or on mzcloud match (level 2). Each match was manually inspected.

For lipidomics, samples were dried under nitrogen flow and resuspended in 60 µl of chloroform. Each sample was split into 2 equal aliquots, one for each polarity analysis. LC-MS analyses were modified from [84] and were performed on an Orbitrap Exactive plus MS (Thermo

Fisher Scientific) in line with an Ultimate 3000 LC (Thermo Fisher Scientific). Each sample was analyzed in positive and negative modes, in top 5 automatic data-dependent MS/MS mode. Column hardware consisted of a Biobond C4 column ($4.6 \times 50$ mm, 5 μm, Dikma Technologies). Flow rate was set to 100 μl min$^{-1}$ for 5 min with 0% mobile phase B (MPB), then switched to 400 μl min$^{-1}$ for 50 min, with a linear gradient of MPB from 20% to 100%. The column was then washed at 500 μl min$^{-1}$ for 8 min at 100% MPB before being re-equilibrated for 7 min at 0% MPB and 500 μl min$^{-1}$. For positive mode runs, buffers consisted for mobile phase A (MPA) of 5 mM ammonium formate, 0.1% formic acid, and 5% methanol in water, and for MPB of 5 mM ammonium formate, 0.1% formic acid, 5% water, 35% methanol in isopropanol. For negative runs, buffers consisted for MPA of 0.03% ammonium hydroxide, 5% methanol in water, and for MPB of 0.03% ammonium hydroxide, 5% water, 35% methanol in isopropanol. Lipids were identified and their signal integrated using the Lipidsearch software (version 4.2.27, Mitsui Knowledge Industry, University of Tokyo). Integrations and peak quality were curated manually before exporting and analyzing the data in Microsoft Excel. Heatmaps and principal component analysis plot were generated using MetaboAnalyst 5.0 [85].

## Egg development and fertility assays

To determine the number of eggs laid and fertility after dsRNA injections or orlistat treatment, treated females were put into individual oviposition cups containing distilled water and Whatman filter paper at 48 to 72 h PBM. One week after oviposition, the number of eggs laid and the fertility was determined using a Nikon SMZ1000 dissecting microscope. Specifically, fertility was assessed by the degree of melanization of laid eggs (black for fertile and brown for infertile), followed by needle pricking of unhatched eggs to assess the presence (or absence) of fully developed larvae.

## Knockdown of gene expression using dsRNA

**Creation of templates for dsRNA production.**   cDNA from female whole bodies without heads was used for RT-qPCR amplification of fragments of interest. Ag*TL2* (*AGAP000211*, 406 bp) and Ag*LSD1* (*AGAP002890*, 485 bp) fragments were amplified using the following primers with the T7 promoter sequence added to their 5′ ends: AgTL2-Fwd: 5′-taatacgactcactatagggCTTTAGTCGGGTGGAGGACA-3′, AgTL2-Rev: 5′-taatacgactcactatagggGATGATCGGTGTTCGGCTT-3′; AgLSD1-Fwd: 5′-taatacgactcactatagggCGCAGATGGAGTCGCTGT-3′, AgLSD1-Rev: 5′-taatacgactcactatagggGCACGAGTCTGGCTGATCTT-3′. GFP fragments were amplified from plasmid pCR2.1-eGFP plasmid as in [48]. The PCR products were confirmed by gel electrophoresis and dsRNA was transcribed using the Megascript T7 transcription kit (Thermo Fisher Scientific) previously described [86,87].

**dsRNA injections.**   For single knockdown experiments, 690 ng of dsRNA (ds*GFP*, dsAg*TL2*, dsAg*LSD1*) was injected (Nanoject III, Drummond) into 1-day-old females at a concentration of 10 ng/nl. Gene knockdown levels were determined in at least 3 biological replicates by quantitative real-time PCR (RT-qPCR).

**RNA extraction, cDNA synthesis, and RT-qPCR.**   For RNA extraction, 7 to 10 female mosquitoes were decapitated in 1× PBS and transferred to Tri-reagent (Thermo Fisher Scientific). Samples were stored at −80°C until the day of extraction. RNA was extracted according to the manufacturer's instructions, with minor modifications. RNA pellet was washed 3 times with 75% ethanol in RNase-free water and air-dried for 7 to 8 min. Pellets were resuspended in heated water for 10 min and DNase-treated using Turbo DNase according to manufacturer's instructions (Thermo Fisher Scientific). Purified RNA was quantified using a NanoDrop

Spectrophotometer 2000c (Thermo Fisher Scientific) and cDNA synthesis was performed as in [87] using 2 μg of RNA in 100 μl cDNA reactions.

For quantification of target genes, RT-qPCR was performed in duplicate either in 7.5 or 15 μl reactions containing 1× PowerUp SYBR Green Master mix for RT-qPCR (Thermo Fisher Scientific), primers, and 2.5 or 5 μl of 10-fold diluted cDNA. Reactions were run on a QuantStudio 6 Pro real time-PCR thermocycler system (Thermo Fisher Scientific). Primers used for RT-qPCR had been previously used or designed spanning exon-exon junctions using Primer-BLAST [88]. Relative quantities of each gene in 3 or more biological replicates were determined using a standard curve with normalization against the ribosomal gene *Rpl19* (*AGAP004422*). With the exception of RT-qPCR primers for Ag*TL2* that have previously been published [48], the following additional primer sets were used: Ag*LSD1*: Fwd: 5′-CATTCAGCATACCTACGAC-3′, Rev: 5′-CACTGGTGGTGATCTTTT-3′; Ag*FAS*: Fwd: 5′-GAGCTTGCGTAGAGATAGAT-3′, Rev: 5′-GCTAAGATTCTGTTGGAATG-3′; *AGAP012959*: Fwd: 5′-CTACTCTGTTCTTCCTGG TC-3′, Rev: 5′-TACCCTGTACTTCGGACTT-3′; *AGAP013072*: Fwd: 5′-TGAAGGCTACTCTG TTCTTC-3′, Rev: 5′-CCTGTACTTCGGACTTTC-3′; *Hsp70* (*AGAP004581*): Fwd: 5′-CGTAC TGTTTCAGTCTCAAG-3′, Rev: 5′-ATCGTACTCTTCCTTCTCTG-3′.

## RNA sequencing and differential gene expression analysis

For RNAseq, RNA was extracted from embryos collected at different time points after oviposition from control and AgTL2-depleted blood fed *An. gambiae* females in 4 biological replicates as above, and then purified using PureLink RNA purification mini kit according to manufacturer's instructions (Invitrogen). Libraries were prepared using a SciClone G3 NGSx workstation (Perkin Elmer) using the Kapa mRNA HyperPrep kit (Roche Applied Science). Polyadenylated mRNAs were captured using oligo-dT-conjugated magnetic beads (Kapa mRNA HyperPrep kit, Roche Sequencing) from 300 ng of total RNA on a Perkin Elmer Sci-Clone G3 NGSx automated workstation. Poly-adenylated mRNA samples were immediately fragmented to 300 to 400 bp using heat and magnesium. First-strand synthesis was completed using random priming followed by second-strand synthesis and A-tailing. A dUTP was incorporated into the second strand to allow strand-specific sequencing of the library. Libraries were enriched and indexed using 11 cycles of amplification (Kapa mRNA HyperPrep kit, Roche Sequencing) with PCR primers, which included dual 8 bp index sequences to allow for multiplexing (IDT for Illumina unique dual 8 bp indexes). Excess PCR reagents were removed through magnetic bead-based cleanup using Kapa Pure magnetic beads on a SciClone G3 NGSx workstation (Perkin Elmer). The resulting libraries were assessed using a 4200 TapeStation (Agilent Technologies) and quantified by qPCR (Roche Sequencing). Libraries were pooled and sequenced on one lane of an Illumina NovaSeq S4 flow cell using paired-end, 100 bp reads.

Sequencing reads were aligned to the *An. gambiae* genome (PEST strain, version 4.14, VectorBase release 62) using HISAT2 (version 2.2.1) with the default parameters. Reads with mapping quality scores <30 were removed using Samtools (version 1.17). The numbers of reads mapped to genes were counted using htseq-count (version 2.0.2) with the default parameters. Calculation of normalized read counts and analysis of differential gene expression was performed using the DESeq2 package (version 1.40.2) in R (version 4.3.1).

## Orlistat treatment

Approximately 5.0 to 6.7 mg orlistat (Sigma-Aldrich) was dissolved in 50 μl pure dimethyl sulfoxide (DMSO) to give a 200 to 270 mM stock solution, which was then diluted as required with 1× PBS to a 10 mM working solution, resulting in 3.7% to 5.0% DMSO in 1× PBS. The 5

mM and 2.5 mM orlistat solutions were prepared by 2-fold serial dilution with 1× PBS. Mated wild-type females were cooled on ice and pre-injected with control or diluted amounts of orlistat (1.0, 0.5, 0.25 μg/mosquito). The amount of orlistat injected was achieved by injecting 207 nl of either the 10, 5, or 2.5 mM solutions, respectively. The control group was injected with the same DMSO concentration as the 10 mM orlistat group (i.e., 3.7% to 5.0% DMSO in 1× PBS). After 3 to 4 h of recovery at 27˚C, some mosquitoes were blood fed and ovaries, midgut, and fat body tissues were dissected from non-blood fed and blood fed females at 24 and 48 h after feeding for TAG analysis. For neutral lipid staining, tissues dissected at 48 h PBM were fixed and stained with the neutral lipid dye LD540. At 3 d PBM, the remaining blood fed females were put into individual oviposition cups to assess the number of eggs developed and fertility. For tarsal delivery, mated wild-type females were allowed to rest on their tarsi for 1 h on glass plates previously coated with orlistat at 1 or 2 mmol/m$^2$ or control plates treated with the acetone vehicle alone. After 30 min of recovery, mosquitoes were blood fed and egg development, and fertility were determined after 7 d of oviposition.

## *P. falciparum* infections of *An. gambiae* mosquitoes

Treated mosquitoes aged 4 to 6 d were housed in small cages and provided about 400 μl of *P. falciparum* gametocyte cultures for 1 h through heated membrane feeders. After blood feeding, poorly fed mosquitoes were removed and cages were maintained for the duration of the experiment in a glove box: 7 to 8 d post infectious feed for oocyst quantification or 13 d for sporozoite quantification. While in the glove box, mosquitoes were provided 10% glucose solution ad libitum.

## Statistical analysis

Data were analyzed using GraphPad Prism 10 and JMP 16 Pro statistical software. GraphPad Prism 10 was used to calculate Fisher's exact test, Student's *t* tests, and 1- or 2-way analysis of variance (ANOVA) with Dunnett's or Šídák's multiple comparisons correction. Prior to ANOVA, data were first assessed for normal distribution and were transformed if found not be normally distributed. JMP Pro 16 was used to generate Least Square Means models incorporating treatment, time, and their interactions as fixed effects and replicate as a random effect. A post hoc *t* test between groups was done to assess significance and *p*-values corrected using a false discovery rate (FDR) of 0.05.

## Supporting information

**S1 Fig. Gene silencing efficacy and lipid staining in AgTL2-depleted and control females after blood feeding.** (A) Ag*TL2* is significantly induced in the fat body but not the ovaries or midgut of females at 2 time points after blood feeding (hPBM = hours post blood meal) compared to non-blood fed (NBF) controls. (Ordinary 2-way ANOVA, Šídák's multiple comparisons correction) (B) Ag*TL2* levels after blood feeding are significantly reduced upon RNAi injections in the fat body but not the ovaries or midgut (Ordinary 2-way ANOVA, Šídák's multiple comparisons correction). (C) Ag*LSD1* levels are reduced upon RNAi injections at 3 d post injection in whole body minus head samples (Rest of body) (Unpaired *t* test). Each dot in A–C represents RT-qPCR analysis from a pool of 10 mosquitoes per replicate (3–5 biological replicates). (D) Neutral lipid staining with LD540 (red) shows intense staining in midgut and fat body tissues and weak staining in ovaries from AgTL2-depleted females at 48 hPBM (scale bar = 10 μm for midguts, 50 μm for ovaries and fat body images). Blue = DAPI (DNA). Numerical data supporting this figure is available in the Harvard Dataverse online repository at https://doi.org/10.7910/DVN/ULTW1K.
(TIF)

**S2 Fig. dsAg*TL2* embryos have altered lipid profiles.** (A) Heatmap of major lipids analyzed by LC-MS reveals dysregulated levels in dsAg*TL2* embryos compared to controls throughout development. (B) Lysophosphatidylethanolamine (LPE) levels are significantly increased in dsAg*TL2* embryos (Ordinary 2-way ANOVA, Šídák's multiple comparisons correction, * = $p < 0.05$). (C, D) Heatmap of acyl composition of (C) top 20 TAGs and (D) 3 cholesterol esters. Heatmaps: values of *t* test statistic, range: blue to red = significant decrease to increase. Three biological replicates, represented by triplicate columns at each timepoint. See S1 Table and S1 Data. Numerical data supporting this figure is available in the Harvard Dataverse online repository at https://doi.org/10.7910/DVN/ULTW1K.
(TIF)

**S3 Fig. Key metabolites are significantly altered in embryos from AgTL2-depleted mothers.** LC-MS analysis reveals dysregulated levels of key metabolites in embryos from AgTL2-deficient females at one or multiple time points after oviposition. Statistical significance was assessed by Least Square Means models testing effect of treatment, time points, and replicate. Adjusted *p*-values were calculated using an FDR = 0.05. Not all significant comparisons are shown for clarity. See S2 Data. (ns = not significant, * = $p < 0.05$, ** = $p < 0.01$, *** = $p < 0.001$, **** = $p < 0.0001$) Numerical data supporting this figure is available in the Harvard Dataverse online repository at https://doi.org/10.7910/DVN/ULTW1K.
(TIF)

**S4 Fig. Gene ontology of dysregulated genes and validation by RT-qPCR.** (A) At 38 h post oviposition, genes involved in neuropeptide signaling pathway, sensory perception, and nervous system processes are most significantly represented among down-regulated genes in dsAg*TL2* embryos compared to controls, while genes involved in protein translation, protein folding, and metabolic processes are most significantly represented in up-regulated genes (numbers in brackets indicate the number of genes in each data set associated with a biological process). (B) RT-qPCR analyses confirm significantly down-regulation of 2 Cchamide-1-related genes (neuropeptide signaling pathway) at 38 h post oviposition in dsAg*TL2* embryos compared to controls, while *Heat shock protein 70* (*Hsp70*) (protein folding) and *fatty acid synthase* (*FAS*) (metabolic process) are significantly up-regulated. Kruskal–Wallis test with Dunn's multiple comparisons correction. See S3 Data. *n* = number of pools of at least 50 embryos per female from 2 biological replicates. Numerical data supporting this figure is available in the Harvard Dataverse online repository at https://doi.org/10.7910/DVN/ULTW1K.
(TIF)

**S5 Fig. Overall representation of metabolic pathways affected in embryos from Ag*TL2*-silenced mothers, based on lipidomics, metabolomics, and transcriptomics analyses.** dsAg*TL2* embryos have significantly reduced levels of main insect sugars and the glycolytic intermediate fructose-bisphosphate, reduced levels of early TCA and ß-oxidation intermediates citrate, aconitate and Acetyl L-carnitine, altered levels of amino acids, and increased levels of nucleotide degradation products (xanthine, hypoxanthine, uric acid, and 3-ureidopropionic acid). Only data from the 38 h time point are shown (but Least Square Means models testing effect of treatment, time point, and replicate were built on the entire time course). Metabolites highlighted in blue are significantly reduced and those highlighted in red are significantly increased. Adjusted *p*-values were calculated using an FDR = 0.05. Not all significant comparisons are shown, for clarity. See S3 Fig and S2 Data. (* = $p < 0.05$, ** = $p < 0.01$, *** = $p < 0.0001$, **** = $p < 0.0001$). Numerical data supporting this figure is available in the Harvard Dataverse online repository at https://doi.org/10.7910/DVN/ULTW1K.
(TIF)

**S6 Fig. Ag*LSD1* accelerates parasite growth and orlistat treatment does not affect lipid levels in fat body.** (A–C) Ag*LSD1* knockdown has (A) no effect on *P. falciparum* infection intensity (Mann–Whitney) and prevalence (P) (Pie charts, Fisher's exact, ns; $p > 0.05$) compared to ds*GFP* controls but leads to (B) increased oocyst size on 7 d PBM and (C) higher sporozoite numbers in salivary glands at 12–13 d PBM (Unpaired *t* test). (D) Orlistat treatment leads to increased glyceride levels in midguts and decreased levels in ovaries compared to control-injected mosquitoes, but no changes are observed in the fat body. All data at 48 h PBM (Ordinary 1-way ANOVA, Dunnett's multiple comparisons correction). Three–four biological replicates are represented in A–C, and 4 in D. *n* = number of individual mosquitoes analyzed (except for glyceride assay, where *n* represents the number of pools of 3 insects). Numerical data supporting this figure is available in the Harvard Dataverse online repository at https://doi.org/10.7910/DVN/ULTW1K.
(TIF)

**S1 Table. Glossary of abbreviations.**
(DOCX)

**S1 Data. Major lipid levels in embryos from control and AgTL2-depleted females at different time points after oviposition.** Lipid levels (as peak area) determined after LC-MS of control and AgTL2-depleted embryos at 4–6 h, 22–24 h, and 36–38 h after oviposition. Values are given for each class and subclass of lipid by fatty acid composition. Total values are summed under each major lipid class in bold. The raw lipidomics data are available from the EMBL-EBI Metabolights online repository using the link https://www.ebi.ac.uk/metabolights/MTBLS9881.
(XLSX)

**S2 Data. Intensities of metabolites determined by LC-MS in embryos from control and AgTL2-depleted mothers at different time points after oviposition.** Metabolite levels (as peak area) determined after LC-MS of control and AgTL2-depleted embryos at 4–6 h, 22–24 h, and 36–38 h after oviposition. Values are given for each metabolite and confidence of metabolite identification (ID). The raw metabolomics data are available from the EMBL-EBI Metabolights online repository using the link https://www.ebi.ac.uk/metabolights/MTBLS9881.
(XLSX)

**S3 Data. RNAseq data sets for embryos derived from control and AgTL2-depleted *An. gambiae* mosquitoes at different time points after oviposition.** Transcriptomics of control and AgTL2-depleted embryos at 4–6 h, 22–24 h, and 36–38 h after oviposition. Each time point has 3 associated spreadsheets: all genes, up-regulated genes and down-regulated genes. Values are given for mean ± SE $\log_2$ (fold change on TL2 depletion), test statistic, and unadjusted and adjusted *p*-values, with gene description and name if available. The raw RNAseq data are available from the GEO database using the link https://www.ncbi.nlm.nih.gov/geo/query/acc.cgi?acc=GSE263712.
(XLSX)

## Acknowledgments

We are particularly grateful to Aaron Stanton, Emily Selland, and Elizabeth Nelson for mosquito rearing and all the members of the Catteruccia laboratory for helpful suggestions. We are

also grateful to The Bauer Core Facility at Harvard University for RNA sequencing and the Harvard Medical School Electron Microscopy Facility for TEM.

The findings and conclusions within this publication are those of the authors and do not reflect the policies or positions of the HHMI or the NIH.

## Author Contributions

**Conceptualization:** Maurice A. Itoe, Flaminia Catteruccia.

**Formal analysis:** Maurice A. Itoe, W. Robert Shaw, Iryna Stryapunina, Charles Vidoudez, Duo Peng, Esrah W. Du, Tasneem A. Rinvee, Yan Yan.

**Funding acquisition:** Maurice A. Itoe, Flaminia Catteruccia.

**Investigation:** Maurice A. Itoe, W. Robert Shaw, Iryna Stryapunina, Charles Vidoudez, Duo Peng, Esrah W. Du, Tasneem A. Rinvee, Naresh Singh, Yan Yan, Oleksandr Hulai, Kate E. Thornburg.

**Methodology:** Maurice A. Itoe, Flaminia Catteruccia.

**Supervision:** Flaminia Catteruccia.

**Visualization:** Maurice A. Itoe, Flaminia Catteruccia.

**Writing – original draft:** Maurice A. Itoe, Flaminia Catteruccia.

**Writing – review & editing:** Maurice A. Itoe, W. Robert Shaw, Iryna Stryapunina, Charles Vidoudez, Flaminia Catteruccia.

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
