## [Editor Report · Decision Letter 0]

12 May 2024

Dear Dr Catteruccia, 

Thank you for submitting your manuscript entitled "Maternal lipid mobilization is essential for embryonic development in a key malaria vector" for consideration as a Research Article by PLOS Biology.

Your manuscript has now been evaluated by the PLOS Biology editorial staff as well as by an academic editor with relevant expertise and I am writing to let you know that we would like to send your submission out for external peer review. However, we would like to consider the manuscript as a Short Report, thus please select that article type from the dropdown menu when you submit the metadata (see below).

Before we can send your manuscript to reviewers, we need you to complete your submission by providing the metadata that is required for full assessment. To this end, please login to Editorial Manager where you will find the paper in the 'Submissions Needing Revisions' folder on your homepage. Please click 'Revise Submission' from the Action Links and complete all additional questions in the submission questionnaire.

Once your full submission is complete, your paper will undergo a series of checks in preparation for peer review. After your manuscript has passed the checks it will be sent out for review. To provide the metadata for your submission, please Login to Editorial Manager (https://www.editorialmanager.com/pbiology) within two working days, i.e. by May 14 2024 11:59PM.

Kind regards,

Ines

--

Ines Alvarez-Garcia, PhD

Senior Editor

PLOS Biology

---

## [Decision Letter · Decision Letter 1]

9 Aug 2024

Dear Dr Catteruccia,

Thank you for your patience while your manuscript "Maternal lipid mobilization is essential for embryonic development in a key malaria vector" went through peer-review at PLOS Biology. Your manuscript has now been evaluated by the PLOS Biology editors, an Academic Editor with relevant expertise, and by several independent reviewers.

As you will see in the reports, the reviewers are generally positive about your work, but some concerns need to be addressed before publication. Reviewer #1 emphasizes the need to update the introduction and incorporate previous literature into the discussion, among other points. Reviewer #3 raises concerns about the overinterpretation of certain results and suggests acknowledging the limitations of targeting lipid metabolism to regulate mosquito reproduction. Overall, it's important that you revise the introduction and conclusion, eliminate speculation, and moderate claims about using lipase inhibitors as potential tools for insect management.

IMPORTANT: While the experimental requests from reviewer #1 would strengthen the conclusion of the study, this are not necessary for acceptance. However, the Academic Editor has provided additional comments about what concerns should have special attention. Addressing such concerns is essential for further consideration of your manuscript for publication in PLOS Biology.

In light of the reviews, which you will find at the end of this email, we are pleased to offer you the opportunity to address the [comments/remaining points] from the reviewers in a revision that we anticipate should not take you very long. We will then assess your revised manuscript and your response to the reviewers' comments with our Academic Editor aiming to avoid further rounds of peer-review, although might need to consult with the reviewers, depending on the nature of the revisions.

**IMPORTANT - SUBMITTING YOUR REVISION**

*Resubmission Checklist*

*Published Peer Review*

*PLOS Data Policy*

*Blot and Gel Data Policy*

Sincerely,

Melissa

Melissa Vázquez Hernández, PhD

Associate Editor 

PLOS Biology

on behalf of

Ines

Ines Alvarez-Garcia, PhD

Senior Editor

PLOS Biology

REVIEWERS' COMMENTS

Reviewer #1: 

This paper presents data on the role of lipids in embryonic development of the malaria vector Anopheles gambiae. The study reports three main findings: 1) RNAi knockdown of a TAG lipase (AgTL2) increased TAG stores in the midgut and fat body of adult females after blood feeding while reducing them in eggs, 2) knockdown of AgTL2 reduces TAG stores in eggs while knockdown of AgTL2 and a lipid storage droplet protein (AgLSD1) near fully inhibited egg hatching, and 3) a commercially available lipase inhibitor (Orlistat) also reduced egg hatching when applied to surfaces that adult females contact without effect on malaria parasite development.

The manuscript I was sent indicated it was an R1 (revised) submission. I was not a reviewer of the original submission and was also not given any previous assessments if in fact the paper was previously sent out for review. With this in mind, my assessment of the manuscript is mixed. 

Strengths include an overall well-written text and associated figures that were easy to read. Reported results support the main conclusions that inhibition of lipase activity greatly reduces egg hatching. Results from the Orlistat experiments provide a proof of principle that lipase inhibitors potentially could be used to reduce mosquito fertility. A third strength from the perspective of mosquito biology is that the oogenesis literature has primarily emphasized the regulation of vitellogenin biosynthesis while focusing somewhat less on the importance of lipid packaging into eggs as a requirement for embryo development. This last point is something the authors don't actually mention in the manuscript (see below) but is a strength in the eyes of this reviewer. 

Per guidelines, short reports in PLoS Biology are supposed to be novel, provocative and of general interest. I honestly was not persuaded the results were novel/provocative for three reasons:

1. Background. The authors begin the paper with a succinct summary of lipolysis in mammals followed by noting that insects possess similar machinery. The authors also introduce adipokinetic hormone (AKH) as a lipolysis activator. AKH has analogous functions to glucagon in mammals that the authors don't state but should have for general readers. The papers the authors cite in the Introduction are pretty dated. They also leave out a fairly substantial literature on the topic across insects generally (many references) and in mosquitoes where a number of recent studies have examined lipid storage, mobilization and packaging into oocytes beginning with Briegel's work in the early 2000s to the present (Pinch et al. 2021 as one of several examples). AKH is a key regulator of lipogenesis in insects but published results in mosquitoes also implicate other hormones in regulating lipid metabolism (see Wang et al. 2017; Dou et al. 2023). A number of studies in different insects also report results that yolk, which primarily consists of protein and lipid, is essential for embryo development. This literature goes well beyond Attardo et al. 2012 and Yang et al. 2024 the authors cite in the Discussion. My reason for making these points is two-fold. First, the Introduction and Discussion leave the impression the lipid storage/mobilization literature in mosquitoes and other insects is thinner than it is. This may have been unintentional on the part of the authors but also left the impression with this reviewer that the paper addressed an area that is less studied than is actually the case. I fully recognize A. gambiae as a malaria vector has applied justifications for reporting interventions affecting disease transmission and that reported findings are new for this species. More fundamentally though, I just didn't see results reporting that macronutrients like lipid are yolk components that embryos require for development as being novel or provocative.

2. Multiomic data show that a lot of processes change with reduced lipid stores in eggs but these findings are overall hardly surprising given the greater literature in insects, other invertebrates, or vertebrates where externally laid eggs have long been known to depend on packaged nutrients for development. The multiomic data are thus voluminous but I also came away concluding this information didn't substantially clarify underlying mechanisms for failure of eggs to hatch because so many things go wrong. A cursory inspection of the mammalian (human) literature makes this emphatically clear where a huge number of defects have been characterized in association with lipid deficiencies. Emphasizing effects on neuronal functions in the Discussion read too much like cherry picking and didn't square with the wider-ranging alterations reported in the results. Lastly, I struggled to see how findings are likely to inform studies on embryo development where maternal nutrients are provisioned by other mechanisms that don't involve yolk formation. Ironically, PLoS Biology published a paper many years ago (Brawand et al. 2008) that looked at how loss of yolk genes in mammals parallel the transition to producing a placenta and other novel machinery (lactation) that provision lipid and other nutrients to embryos. 

As noted above, the mosquito literature has primarily focused on packaging of vitellogenin into eggs, while a novel feature of this study is its focus on the consequences of reducing lipid stores in oocytes. I would hypothesize a key reason for this is that vitellogenin is actually a precursor for the lipo- and phosphoproteins that are the predominant components of yolk in all animals that lay eggs. Since the authors make no mention of vitellogenin in the manuscript, I would recommend they consider doing so in the context of reported results.

3. Reporting that Orlistat can be topically delivered to A. gambiae and reduce egg hatching is a new result for any insect. I would be remiss though to not mention the approach near fully follows other studies the authors have published where drugs were added to bednets that reduced parasite loads in A. gambiae. This again goes to novelty as results follow previous approaches quite closely. I also noticed the authors used Orlistat at a dose of 1 mmol/m2 to reduce fertility which at retail costs works out to be about $20/m2 which is massively higher than any currently used insecticide.

Points that should be revised if the paper moves forward in PLoS Biology or another journal.

1. Fig. 1B. Unclear from the methods or figure legend what is being shown. Are these oviposited eggs and the Y axis is showing number of eggs laid per GFP or TL2 treated female? If yes, how was it discerned the eggs developed versus the eggs were simply laid? Same questions apply to Fig. 1F, Fig. 4B. Thus, please provide clarification for what egg development refers to.

2. Orlistat treatment. The authors should state in the methods whether Orlistat was dissolved in DMSO and that 207 nl of Orlistat in pure DMSO was injected into females. I raise this point because many insects respond extremely poorly to injection of even small amounts of DMSO into the hemocoel.

3. This study did not examine the underlying mechanisms regulating lipolysis in A. gambiae which in the mosquito literature has primarily been studied in A. aegypti. This is fully fine given short report guidelines for PLoS Biology. However, the older literature also provides hints oogenesis may not be fully conserved between culicine and anopheline mosquitoes. Given egg formation is activated by blood which contains lipid as well, it may be more significant than noted that TAG levels in the midgut were more strongly affected by TL2 knockdown than in the fat body (Fig. 1C, D). Along these lines, Fig. S1 reports AgTL2 knockdown efficacy in the fat body and 'rest of body' but not in the gut or ovaries, which could also have roles in reducing TAG stores available to embryos. It also would be valuable to know if knockdown of the genes the authors targeted actually alter yolk deposition into oocytes which is not apparent from reporting lower TAG levels in whole ovaries, egg laying data, or the morphological data presented in Fig. 2 which only shows late stage (post-gastrulation) embryos. All this to say the authors are assuming phenotypes are due to suppressed lipolysis in the fat body which may not fully be the case given the results presented.

Reviewer #2: 

This is a 'short report' by Itoe and coworkers on lipid mobilization in female Anopheles gambiae. They knocked down two genes - TL2 and LSD1 - a lipase and a Lipid storage droplet protein and observed strong distinct phenotypes in egg and embryo development and Plasmodium development using lipidomics, metabolomics, and RNAseq. This is a really nice study and the manuscript is in very good shape. I have really only some minor comments. 

Line 91/148 'impared neuronal function' is pure speculation. Show evidence or take it out...

Line 173 to 179 I understand the motivation of the authors to present lipase inhibitors as potential future insect management tools. However, there is virtually no chance that this will work out. Mosquito control is about killing mosquitoes and not sterilizing them, with the exception of sterile insect technique. I recommend to remove this language. 

Line 423 Did the experimenters really use field collected mosquitoes for their experiments? That is hard to believe... Please give a more detailed description of their origin. How could anyone repeat these experiments when no established laboratory strain was used?

Line 549 How do you determine fertility with a stereo microscope? It is impossible to distinguish dead and living eggs. Please explain. 

Supplemental Figure 1 C. Check labels.

Reviewer #3: 

The manuscript "Maternal lipid mobilization is essential for embryonic development in a key malaria vector" presents significant findings on the role of lipid mobilization in the reproductive biology of Anopheles gambiae. The research demonstrates that impairing lipolysis in female mosquitoes leads to embryonic lethality, providing insights into potential new strategies for mosquito control. The paper is well written and the study is well-conducted, the data are robust and the figures are clear and easy to interpret. However, there are a few areas where the manuscript could be strengthened in terms of interpretation of the embryonic neuronal phenotype and potential challenges associated with implementing a mosquito control solution based on inhibition of lipid metabolism.

Some conclusions, such as attributing all developmental issues to impaired neuronal functions, may be overinterpretations. It would be good to highlight that potentially multiple factors could be involved and suggest areas for further research. The discussion on how lipid deficiencies affect neuronal development and function should be expanded to include possibilities beyond energy metabolism as the reason for nervous system disruption. Mention the various roles of triglycerides and their derivatives in the nervous system, such as membrane synthesis/synapse function, production of key signaling molecules from fatty acids and cholesterols, and myelination.

I think something else that would be useful is consideration of the potential issues and difficulties that would arise from targeting lipid metabolism given how conserved these systems are across biology. It is important to discuss the potential ecological impact of targeting lipolysis, particularly concerning non-target organisms in addition to the safety concerns of using lipase inhibitors in mosquito control, especially regarding potential effects on humans who handle and sleep in close contact with these bednets. Given the conserved nature of lipolysis, this is a significant concern that needs to be discussed. A more detailed examination of this risk would enhance the paper's credibility by demonstrating consideration of broader ecological consequences. It is mentioned to generate an Anopheles specific inhibitor, but it would be good to expand upon the research required and challenges associated with the production of such an intervention.

Also, make sure all the abbreviations are defined at first use and consider adding a glossary of abbreviations at the beginning of the manuscript for the benefit of the reader.

With these revisions, I believe the manuscript will make a substantial contribution to the field. The findings are significant and could potentially inform new strategies for controlling malaria vectors. I recommend acceptance with minor revisions.

ACADEMIC EDITOR'S COMMENTS:

- revise the introduction and discussion sections to better reflect the current literature on the topic

- include caveats regarding the interpretation of multiomic data

- remove speculation about causative relationship between developmental issues and impaired neuronal functions

- tone down the potential of lipase inhibitors as potential tools for insect management, due to both conceptual (sterilization vs. killing) and practical (cost, potential off-target effects) limitations.

---

## [Editor Report · Decision Letter 2]

18 Oct 2024

Dear Dr Catteruccia,

Thank you for your patience while we considered your revised manuscript entitled "Maternal lipid mobilization is essential for embryonic development in a key malaria vector" for publication as a Short Report at PLOS Biology. This revised version of your manuscript has been evaluated by the PLOS Biology editors and by the Academic Editor;

Based on our Academic Editor's assessment of your revision, we are likely to accept this manuscript for publication, provided you satisfactorily address the data and other policy-related requests stated below.

In addition, we would like you to consider a suggestion to improve the title:

""Maternal lipid mobilization is essential for embryonic development in the malaria vector Anopheles gambiae"

We expect to receive your revised manuscript within two weeks. 

*Published Peer Review History*

*Press*

Sincerely,

Ines

--

Ines Alvarez-Garcia, PhD

Senior Editor

PLOS Biology

Fig. 1B-G; Fig. 3A-D; Fig. 4B, C, E-G; Fig. S1A-C; Fig. S2A-D; Fig. S3; Fig. S4A, B; Fig. S5 and Fig. S6A-D

**In addition, please make sure that the data you deposited in EMBL-EBI MetaboLights online database and in GEO are made publicly available at this stage.

CODE POLICY

---

## [Editor Report · Decision Letter 3]

29 Nov 2024

Dear Dr Catteruccia,

Thank you for the submission of your revised Short Report entitled "Maternal lipid mobilization is essential for embryonic development in the malaria vector Anopheles gambiae" for publication in PLOS Biology. On behalf of my colleagues and the Academic Editor, Louis Lambrechts, I am delighted to let you know that we can in principle accept your manuscript for publication, provided you address any remaining formatting and reporting issues. These will be detailed in an email you should receive within 2-3 business days from our colleagues in the journal operations team; no action is required from you until then. Please note that we will not be able to formally accept your manuscript and schedule it for publication until you have completed any requested changes.

PRESS

Sincerely, 

Ines

--

Ines Alvarez-Garcia, PhD

Senior Editor

PLOS Biology
